# Enhance the Surface Insulation Properties of EP Materials via Plasma and Fluorine-Containing Coupling Agent Co-Fluorinated Graphene

**DOI:** 10.3390/nano14242009

**Published:** 2024-12-14

**Authors:** Manling Dong, Zhifei Yang, Guowei Xia, Jiatao Zhang, Zhenyu Zhan, Weifeng Xin, Qilin Wang, Bobin Xu, Yujin Zhang, Jun Xie

**Affiliations:** 1State Grid Henan Province Electric Power Corporation Research Institute, Zhengzhou 450099, China; dongmanling@ha.sgcc.com (M.D.); yangzhifei@ha.sgcc.com (Z.Y.); zhenyuzhan@ha.sgcc.com (Z.Z.); xinweifeng@ha.sgcc.com (W.X.); wangql@ha.sgcc.com (Q.W.); 2Hebei Provincial Key Laboratory of Power Transmission Equipment Security Defense, North China Electric Power University, Baoding 071003, China; xiaguoweihd@163.com (G.X.); 15203266738@163.com (B.X.); yj140929190@163.com (Y.Z.); 3State Grid Henan Province Electric Power Company, Zhengzhou 450018, China; zhangjt@ha.sgcc.com

**Keywords:** plasma, coupling agent modification, epoxy resin, graphene nanosheets, insulation property

## Abstract

Epoxy resin (EP) is an outstanding polymer material known for its low cost, ease of preparation, excellent electrical insulation properties, mechanical strength, and chemical stability. It is widely used in high- and ultra-high-voltage power transmission and transformation equipment. However, as voltage levels continue to increase, EP materials are gradually failing to meet the performance demands of operational environments. Thus, the development of high-performance epoxy resin materials has become crucial. In this study, a combined treatment using plasma and a fluorine-containing coupling agent was employed to fluorinate graphene nanosheets (GNSs), resulting in DFGNSs. Different concentrations of GNSs/DFGNS-modified EP composites were prepared, and their effects on enhancing the surface insulation properties were studied. Tests on surface flashover voltage, surface charge dissipation, trap distribution, and surface resistivity demonstrated that both GNSs and DFGNSs significantly improve the insulation properties of EP materials. Optimal improvement was achieved with a DFGNS content of 0.2 wt%, where the flashover voltage increased by 16.23%.

## 1. Introduction

Epoxy resin (EP) is an excellent polymer material known for its low cost, ease of production, superior electrical insulation, mechanical strength, and chemical stability. As a result, it is widely used across various industries, including construction, aerospace, automotive, electronics, and marine applications. In the electronics and electrical sector, epoxy resin is primarily used in various insulating components of power systems [1,2]. However, as the operating lifespan of insulating equipment increases, factors such as insulation performance, temperature rise, and mechanical stress on electrical devices have also significantly escalated. Consequently, the insulation properties of epoxy resin composites have become a critical factor affecting the safe and stable operation of these devices [3,4]. Therefore, developing epoxy resin composites with superior performance at a low cost is crucial for ensuring the safe and stable operation of power equipment.

Long-term operation of insulating equipment can lead to the accumulation of charge at the gas–solid interface of epoxy composite insulators, causing field distortion near the insulator. This can result in partial insulation failure and surface flashover, which threatens the safe and stable operation of the power grid [5,6]. Studies have shown that adding inorganic nanoparticles can combine the excellent properties of the nanoparticles with the matrix, thereby enhancing the surface withstand voltage and thermodynamic properties of epoxy composites [7,8]. However, most inorganic fillers lack reactive groups on their surfaces, making it difficult to form bonding interactions with the polymer matrix. This results in a significant reduction in the effectiveness of the fillers in enhancing the surface charge dissipation rate of epoxy composites. Surface treatments, such as coupling modification of nanoparticles, can be used to adjust their surface activity and strengthen the interface interaction with the epoxy matrix [9,10]. These methods are crucial for improving the surface withstand voltage and thermodynamic performance of epoxy composites. Plasma fluorination is an advanced material surface treatment technology. Through the interaction between active fluorine substances in plasma and the material surface, fluoro-containing groups are grafted to the surface of the filler to modify the filler, improve the surface compatibility of the filler and epoxy resin, and thus modify the epoxy resin to improve the insulation performance of the epoxy resin [11,12].

Graphene (GN) is a two-dimensional material composed of a single layer of carbon atoms arranged in sp^2^ hybridized orbitals. Since its first isolation from graphite in 2004, it has generated significant global interest due to its remarkable physical and chemical properties [13,14]. As the thinnest and strongest known nanomaterial, graphene is widely used in epoxy composites due to its exceptional electrical conductivity, low permeability, and high thermal conductivity [15,16]. When used as a nanofiller in combination with epoxy resin, graphene can enhance the electrical, thermal, and mechanical properties of the composite material. These advantages have made graphene a key research focus and widely applicable nanofiller [17,18]. However, several studies have shown that after incorporating graphene into epoxy resin, the performance parameters, such as surface pressure resistance, mechanical properties, and thermal stability, do not always meet the expected results. Due to its unique two-dimensional structure, graphene has poor compatibility with the epoxy matrix, leading to aggregation. This aggregation hinders the effective transfer of external charge, heat, and mechanical stress, limiting the overall performance of the graphene/epoxy composites [18,19,20]. There is currently limited research on the modification of graphene and its compatibility with the epoxy matrix. Therefore, it is essential to explore methods for modifying graphene to improve its dispersion within the epoxy matrix and enhance the electrical properties of the graphene/epoxy composites.

On the basis of the above analysis, this study employs a ball milling method to convert graphene (GN) into graphene nanosheets (GNSs). The GNSs are then fluorinated through a plasma-coupling agent-assisted process to produce fluorinated graphene nanosheets (DFGNSs). On the basis of the filler characterization results, along with tests on the surface flashover voltage, charge dissipation, trap distribution characteristics, and surface resistivity of two types of GNSs/EP composite materials, this study investigates the effect of plasma-coupling agent-assisted fluorination of GNSs on the insulation properties of epoxy resin composites.

## 2. Experiments

### 2.1. Material

Isopropanol ((CH_3_)_2_CHOH) was produced by Huihang Company (Tianjin, China). Graphite powder (purity 99.95%) was supplied by Aladdin Reagents Co., Ltd (Shanghai, China). Fluorinated coupling agent: 1H, 1H, 2H, 2H-perfluorodecyltrimethoxysilane (PFDTMS, C_13_H_13_F_17_O_3_Si), 97% purity, was supplied by Aladdin Reagents Co., Ltd (Shanghai, China). Argon (Ar, purity 99.999%) was supplied by Beijing Oxygen Lia Chemical Gas Co., Ltd (Beijing, China). Carbon tetrafluoride (CF₄, purity 99.999%) was supplied by Beijing Oxygen Lia Chemical Gas Co., Ltd (Beijing, China). Release agent: JIA-DAN909A, used for the release of epoxy resin samples, was supplied by Guangdong Jiadan Lubricants Co., Ltd (Guangdong, China). Bisphenol A diglycidyl ether (DGEBA, E-51), methyl tetrahydrophthalic anhydride (MTHPA, 504), and 2,4,6-tris(dimethylaminomethyl)phenol (DMP-30) were all purchased from Shanghai Resin Factory (Shanghai, China).

### 2.2. Fluorination of GNSs and Preparation of EP Composites

Before the fluorination treatment, graphene nanosheets (GNSs) were prepared using a ball milling method (Figure 1) [21,22]. A total of 4 g of graphite powder was dissolved in 100 mL of isopropanol. The mixed solution was placed in a ball mill jar, and the milling speed was set to 300 rpm for a duration of 48 h. After milling, the mixture was placed in a drying oven and heated at 120 °C for 2 h. The dried mixture was then sonicated to ensure thorough dispersion. The solution was filtered using a vacuum pump and filter apparatus to obtain the GNS nanofillers.

For the fluorination process, GNSs were mixed with PFDTMS in a 5:6 mass ratio to form a gel-like mixture, which was then placed in a quartz reaction vessel. Fluorine-containing CF_4_ gas and Ar protective gas were introduced through L-shaped vents at a ratio of 0.4:2.5. A high-frequency power supply was set to an output voltage of 7 kV and a frequency of 50 kHz, and the power supply was activated. The mixture was stirred every 5 min to ensure uniform fluorination, with a total fluorination time of 15 min. After fluorination, the filler was placed in a quartz boat and calcined in a tube furnace at 280 °C for 2 h to remove any residual fluorine-containing coupling agents from the graphene powder. The calcined powder was then ground and sealed in bottles for storage, resulting in plasma-assisted fluorinated graphene with coupling agents, referred to as DFGNSs.

The epoxy resin composites were prepared by mixing DGEBA epoxy resin with MTHPA hardener in a mass ratio of 100:80:1. GNSs and DFGNSs were added as fillers at mass fractions of 0.1%, 0.2%, 0.3%, 0.4%, 0.5%, and 0.6%. The mixture was stirred in a 60 °C oil bath at 200 rpm for 10 min. DMP-30 curing agent was then added, and the mixture was stirred for an additional 10 min. The mixture was placed in a vacuum drying oven at 80 °C, degassed under vacuum for 5 min, and then removed. After spraying the release agent uniformly on the surface of the mold, heating pretreatment was carried out, and the mixed solution was poured into the mold. The drying oven was heated to 120 °C, and the curing process lasted for 3 h. After curing, the samples were removed from the mold and cooled to room temperature, and the resulting composite samples were labeled as GNSs/EP and DFGNSs/EP.

### 2.3. Characterization and Testing

The surface morphology of GN, GNS, and DFGNS fillers was characterized using a scanning electron microscope (SEM). Fourier transform infrared spectroscopy (FTIR) and X-ray photoelectron spectroscopy (XPS) were used to analyze the elemental composition and bonding structures of GNS fillers before and after fluorination.

Isothermal surface potential decay (ISPD) testing was conducted on GNSs/EP and DFGNSs/EP composites to measure surface potential decay [23,24]. The experimental conditions were set with a temperature of 25 °C and a relative humidity of 25%. The 7 kV negative high-voltage DC power supply was used to charge a stainless-steel needle, 10 cm in length with a 0.5 mm radius of curvature, positioned 5 mm above the sample surface for 1 min. After charging, the needle electrode was removed, and an active capacitance probe was positioned 2 mm above the sample surface to measure the potential decay at a sampling frequency of 0.01 kHz. Data acquisition was initiated to record surface potential decay over 1800 s. The potential decay curve was further analyzed to calculate the trap distribution on the material’s surface. The surface potential decay function *f*(*t*) for insulating materials is related to the trap energy level *El* and trap density *De*, as described by the following Equations (1)–(3) [25,26]:(1)$$El=k_{B}T\ln\left(v_{a}t\right)$$
(2)$$De=t\frac{\varepsilon_{r}\varepsilon_{0}}{qI}\frac{df\left(t\right)}{dt}$$
(3)$$f\left(t\right)=Xe^{-t/x}+Ye^{-t/y}$$
where *k_B_* is the Boltzmann constant, *T* is the absolute temperature in Kelvin (K), *v_a_* is the electron escape frequency, *q* is the elementary charge, and *I* is the sample thickness. *X*, *Y*, *x*, and *y* are the fitting parameters of a double exponential function. The trap energy level distribution curve on the composite material’s surface can be obtained from the calculation results.

The surface potential distribution experiment followed the same charging method as the isothermal surface potential decay technique. After charging was complete, the needle electrode was removed, and an active capacitance probe was positioned 2 mm above the sample. The sampling frequency was set to 0.1 kHz. A stepper motor was used to scan the potential distribution around the charging center, with a step size of 1 mm, and the test duration was 90 s [27,28].

The surface flashover test used a finger-type electrode [29,30,31] with a bottom diameter of 4 mm and a total electrode length of 15 mm. The test was conducted in a temperature- and humidity-controlled high-voltage shielded box, with a temperature of 25 °C and humidity at 45%. A uniform voltage boost method was applied to measure the flashover voltage of the GNSs/EP and DFGNSs/EP surfaces, with an electrode gap of 5 mm. A negative polarity DC power supply was used to apply a uniform voltage boost at a rate of 50 V/s until surface flashover occurred. The flashover voltage was recorded by transmitting the instantaneous signal through a high-voltage probe to an oscilloscope. Each sample type was tested 12 times, and the average value was taken as the flashover voltage for the sample.

## 3. Results

### 3.1. Graphene Surface Morphology Changes

Figure 2 shows the SEM images of GN, GNSs, and DFGNSs. As shown in Figure 2a, the unmilled GN exhibits fewer faults, with a stacked and aggregated layer structure. Its surface and edges are relatively smooth. After milling in the ball mill, the GNSs show successful exfoliation into a thin, film-like structure with noticeable bending. However, the surfaces and edges of the GNSs remain smooth. After plasma-fluorine coupling treatment, the surface of the DFGNSs displays distinct granular attachments, with raised features on both the surface and edges, indicating that the plasma-fluorine coupling treatment roughened the DFGNSs.

### 3.2. Graphene Surface Element Changes

Figure 3 shows the FTIR spectra of GNS fillers before and after two types of fluorination treatments. The results indicate that both fillers exhibit a strong absorption peak around 3435 cm^−1^, corresponding to the –OH group attached to the graphene surface. For DFGNSs, peaks at approximately 803 cm^−1^ and 1265 cm^−1^ are observed, corresponding to the C–F and –CHF_2_ groups, respectively, while the remaining wavenumber range shows no significant difference from the untreated spectrum. This suggests that fluorine-containing groups were successfully grafted onto the graphene surface.

XPS analysis was performed on GNS fillers before and after fluorination, with the results shown in Figure 4. Both DFGNSs and GNSs exhibit common graphene absorption peaks: the C1s peak at 285.4 eV, and the O1s peak at 531.8 eV. Additionally, the DFGNSs show a distinct F 1s peak at 689 eV, an Auger peak for fluorine at 829 eV, and characteristic Si 2s and Si 2p peaks at 150 eV and 105 eV, respectively. Peak fitting of the C 1s spectrum provides a clearer understanding of the bonding forms of the elements. Analysis reveals that in GNSs, carbon predominantly exists in the C-C bonding form, while in DFGNSs, additional bonding forms such as C-OH, C-F, and C-F_3_ are present. This indicates that plasma-fluorine coupling treatment effectively grafts a significant number of fluorine-containing groups onto the GNSs’ surface, predominantly in the –CF3 and –CF bonding forms. The FTIR characterization results are in strong agreement with the XPS findings.

### 3.3. Surface Flashover Voltage and Surface Resistance Test Results

Figure 5 shows the surface flashover voltage curves of GNSs/EP composites at different mass fractions before and after fluorination. As shown in the figure, the flashover voltage of pure EP is 8.87 kV. For both GNSs/EP and DFGNSs/EP materials, the surface flashover voltage increases initially and then decreases as the filler concentration increases. The GNSs/EP composite exhibits a peak flashover voltage of 9.77 kV at 0.4 wt%, which is a 10.15% improvement over pure EP. By contrast, DFGNSs/EP reaches its peak flashover voltage of 10.31 kV at a lower doping concentration of 0.2 wt%, representing a 16.23% improvement compared to pure EP. The experimental results demonstrate that both GNSs and DFGNSs doping improve the surface insulating properties of epoxy composites to varying extents, with DFGNSs showing a more significant enhancement.

Figure 6 presents the surface resistivity measurements of GNSs/EP composites before and after fluorination. The surface resistivity of pure epoxy resin is 1.35 × 10^18^ Ω. Doping small amounts of GNSs and DFGNSs increases the surface resistivity of EP composites, enhancing the material’s ability to regulate surface charge and reducing the rate of charge migration. The surface resistivity of the composites exhibits a similar trend to the flashover voltage. The surface resistivity curves for both GNSs/EP composites show a general increase followed by a decrease, with peak values occurring at 0.4 wt%. At this point, the surface resistivity of GNSs/EP is 3.5 × 10^18^ Ω, and that of DFGNSs/EP is 8 × 10^18^ Ω. However, due to the conductive nature of graphene, increasing the doping concentration further reduces the resistivity of the composite material. At a concentration of 0.6 wt%, the surface resistivities of the two composites are reduced to 0.9 × 10^18^ Ω and 0.4 × 10^18^ Ω, respectively, which are lower than that of pure EP. This indicates that graphene can only enhance the insulating properties of EP material at lower doping concentrations. At higher concentrations, however, it has a negative effect on the improvement of insulation performance.

### 3.4. Surface Charge Dissipation and Trap Distribution

This paper converts the surface potential distribution into surface charge decay rates and normalizes the surface potential. The charge decay behavior of GNSs/EP and DFGNSs/EP composites is shown in Figure 7. Compared to pure epoxy resin, the incorporation of GNSs alters the charge decay rate of the composite, demonstrating the relationship between charge dissipation and the filler material. As the doping concentration increases, the surface potential of both EP composites shows a consistent downward trend. The surface potential decay rate of GNSs/EP and DFGNSs/EP initially decreases and then increases. When the doping concentration of GNSs and DFGNSs reaches 0.6%, the surface potential decay rate of both EP composites significantly accelerates.

By calculating the charge decay rate, the distribution of surface traps in the composites can be obtained, providing further insight into the changes in the insulation properties of the EP composites, as shown in Figure 8. It can be observed that both GNSs/EP and DFGNSs/EP composites predominantly feature deep traps. After fluorination, the deep trap energy levels of DFGNS-modified EP materials vary more significantly with concentration. For GNSs/EP composites, the deep trap energy levels are mainly concentrated between 1.0 eV and 1.05 eV, while for DFGNSs/EP composites, they vary within the range of 1.05 eV to 1.10 eV. This is primarily due to the presence of fluorine-containing functional groups grafted on the surface of DFGNSs, which enhance the binding effect on surface charges. Additionally, similar to the results for surface resistivity and charge decay, when the filler concentration reaches 0.6 wt%, the trap energy levels in the EP material rapidly decrease, with a trend toward the formation of shallow traps.

## 4. Discussion

The distortion of the surface electric field can reflect the surface insulation performance of EP composites. This section utilizes the surface potential distribution of GNSs/EP and DFGNSs/EP samples to reveal the mechanisms by which these two fillers enhance the surface insulation properties of EP materials. The test results are shown in Figure 9.

Before the experiment, the samples of different EP composites were charged using a negative polarity corona discharge, with the charging center located at X = 20 mm and Y = 20 mm. As a result, the surface potential test results exhibit a “triangular valley-like” distribution, with a central depression. By comparing the steepness of the valley, we can determine the distribution of surface charges and infer the surface electric field distribution. Figure 9a shows the surface potential distribution of the Pure EP sample. The potential in the charging center is significantly higher than in the surrounding areas. Additionally, the potential contour surrounding the center is relatively steep, indicating that small changes in distance cause significant potential variations. This suggests a higher field distortion rate and a relatively poor surface insulation performance for the EP material. Figure 9b–g show the surface potential distribution of GNSs/EP composites with varying concentrations of GNSs. After doping with GNSs, the potential distribution near the charging center becomes more uniform compared with pure EP. Changes in distance no longer cause large potential variations, resulting in a more gradual electric field change, which improves the surface insulation performance of the EP material. Figure 9h–m show the surface potential distribution of DFGNSs/EP composites with different concentrations of DFGNSs. After doping with DFGNSs, the potential distribution near the charging center becomes even smoother compared with both pure EP and GNSs/EP composites, forming a “trapezoidal valley-like” distribution. Small changes in distance lead to minimal variations in surface potential, resulting in a more uniform electric field distribution and a higher surface insulation strength for the EP material. This trend is consistent with the surface tracking voltage test results of the EP materials. The more uniform the surface potential distribution, the higher the corresponding flashover voltage.

It is believed that when GNSs interact with the epoxy interface, the interface layer forms a potential barrier due to Coulombic blocking, making it difficult for injected negative charges to escape once trapped. However, since the energy levels of the deep traps are not high, the number of charges that can be trapped is limited. As the voltage increases, charges are continuously injected from the electrode to the material surface. A small amount of charge is captured by the deep trap in the EP material near the high-voltage electrode, and the rest of the charge continues to migrate to the earth electrode and is captured by the deep trap in the material in the next region. Due to the limited amount of charge that can be captured by deep traps, the electric field on the material’s surface remains relatively uniform, preventing significant charge accumulation over short periods. This is beneficial for improving the surface insulation properties of the EP composite material. DFGNSs treated with plasma and fluorine-containing coupling agents disrupt the planarity and integrity of graphene while grafting a large number of highly electronegative fluorinated long chains. Additionally, these fluorinated long chains possess higher chemical reactivity, enabling them to form stronger bonds with the EP matrix. These effects lead to an increase in the deep trap energy levels within the EP composite material, enhancing both the ability and the quantity of charges that can be trapped. Consequently, more charges are trapped within the same period, which helps improve the uniformity of the electric field distribution on the material’s surface. However, as the energy levels and density of the deep traps increase, the number of trapped charges also rises, making the concentration of DFGNSs a critical factor in limiting further improvement of the EP material’s surface insulation properties. At a filler concentration of 0.2 wt%, the EP material achieves its highest breakdown voltage. As the concentration increases further, the number of trapped charges in the material’s deep traps rises, leading to greater distortion of the electric field and a subsequent reduction in the EP material’s surface insulation performance. In summary, at low doping concentrations, DFGNSs can effectively enhance the surface insulation strength of EP materials (Figure 10).

## 5. Conclusions

In this study, plasma treatment and coupling agent modification techniques were applied to graphene nanosheets (GNSs), focusing on improving the surface insulation properties of epoxy (EP) composites. By analyzing the surface charge dissipation, trap distribution, and surface potential distribution, the improvement mechanism is discussed. The conclusions are as follows:(1)After ball milling, the graphene nanosheets are no longer multi-layer structures but single-layer structures, and the graphene nanosheets are stripped out of thinner and more irregular nanosheet structures.(2)The results of FTIR and XPS indicate that the surface plasma coupling agent-assisted fluorination treatment successfully grafted fluorine-containing groups onto the surface of GNSs to achieve the functional treatment of GNSs.(3)The experimental results of flashover show that the concentration of GNSs and DFGNSs affects the insulation performance of EP. When a low concentration of GNSs and DFGNSs is added, the surface flashover voltage of EP composite is increased by 16.23%. The enhancement effect of DFGNSs is better than that of GNSs. However, at higher packing concentrations, the insulation properties of EP materials are negatively affected.(4)The trap distribution, surface resistivity, and surface potential distribution of DFGNSs show that the long-chain fluorinated groups on the surface of the two-dimensional nanosheet structure mainly regulate the surface charge of EP composites. This makes the surface potential distribution more uniform in a short time, reduces the electric field distortion rate, and thus significantly improves the surface insulation performance of EP materials.

## Figures and Tables

**Figure 1 nanomaterials-14-02009-f001:**
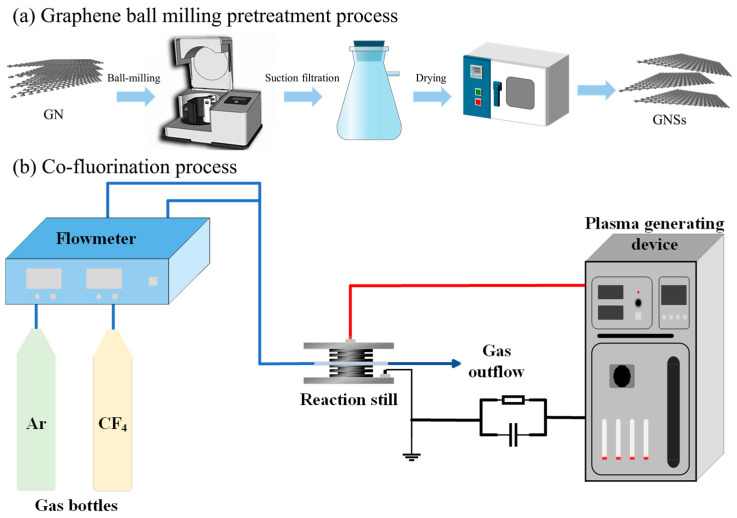
The GN ball milling treatment and GNSs fluorination treatment platform. (**a**) Graphene ball milling pretreatment process; (**b**) Co-fluorination process.

**Figure 2 nanomaterials-14-02009-f002:**
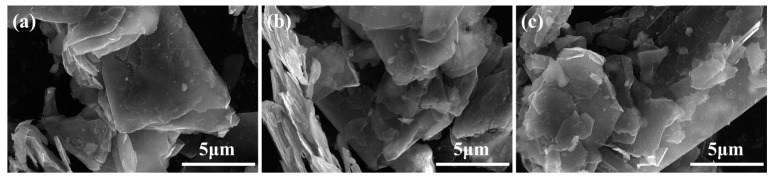
The SEM test results. (**a**) GN; (**b**) GNSs; (**c**) DFGNSs.

**Figure 3 nanomaterials-14-02009-f003:**
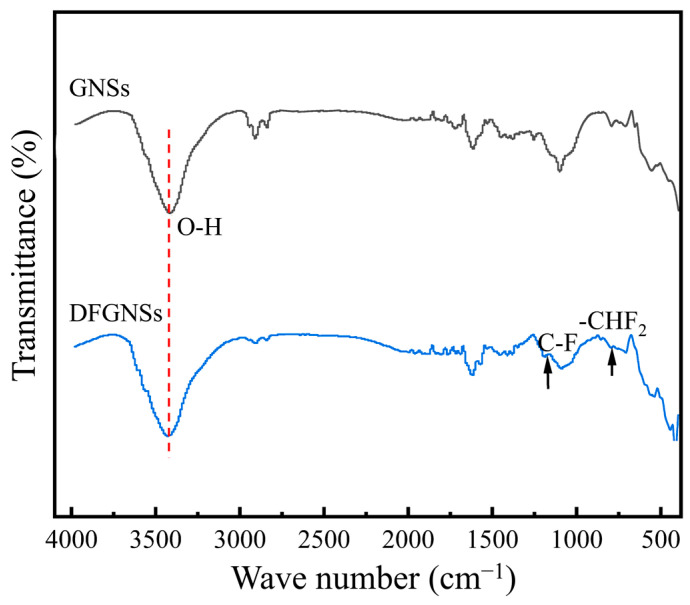
The FTIR results of GNSs before and after fluoridation.

**Figure 4 nanomaterials-14-02009-f004:**
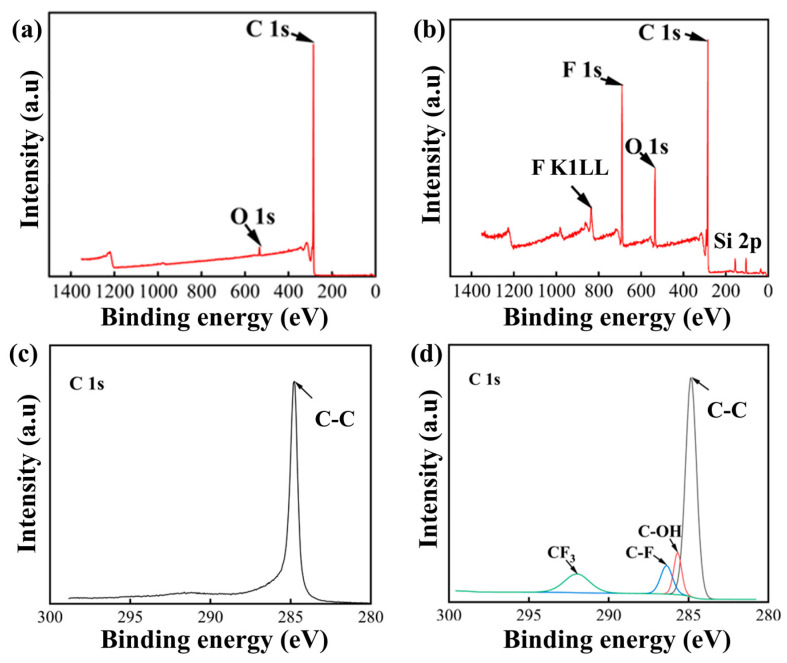
The XPS full spectrum and C1s partial peak spectrum. (**a**) GNSs full spectrum; (**b**) DFGNSs full spectrum; (**c**) C1s partial peak of GNSs; (**d**) C1s partial peak of DFGNSs.

**Figure 5 nanomaterials-14-02009-f005:**
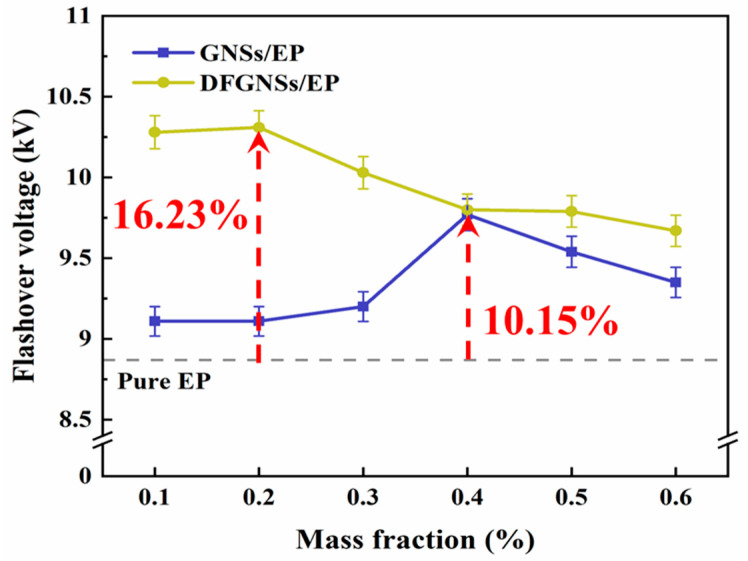
The flashover voltage test results of EP composites.

**Figure 6 nanomaterials-14-02009-f006:**
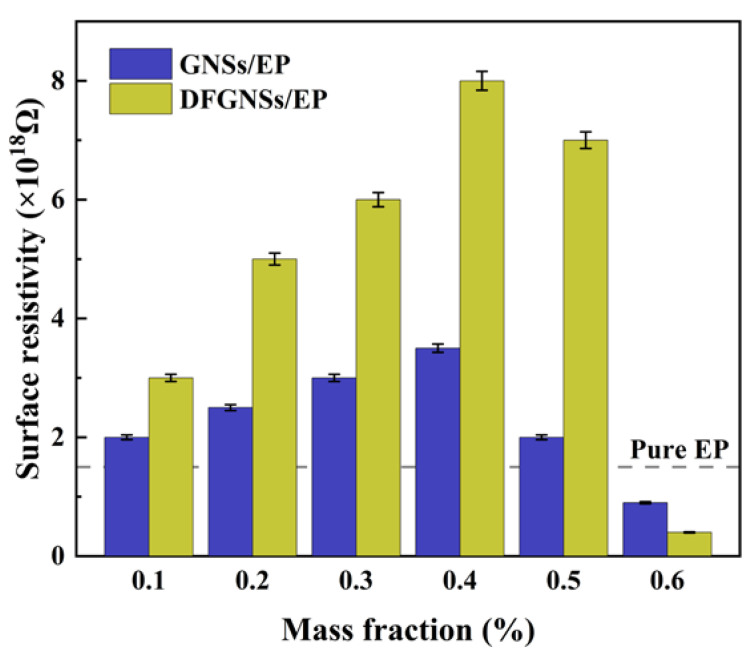
The surface resistivity test results of EP composites.

**Figure 7 nanomaterials-14-02009-f007:**
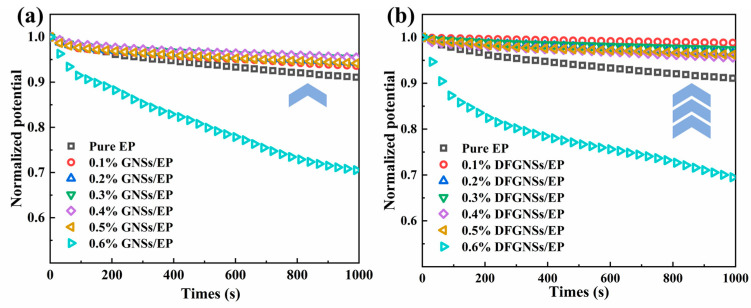
The surface charge dissipation curve of EP composites. (**a**) GNSs/EP; (**b**) DFGNSs/EP.

**Figure 8 nanomaterials-14-02009-f008:**
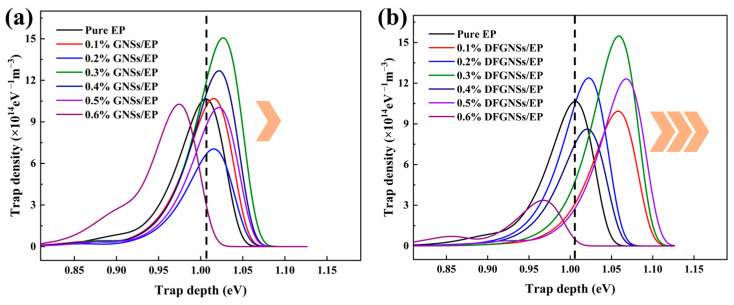
The trap distribution characteristics of EP composites. (**a**) GNSs/EP; (**b**) DFGNSs/EP.

**Figure 9 nanomaterials-14-02009-f009:**
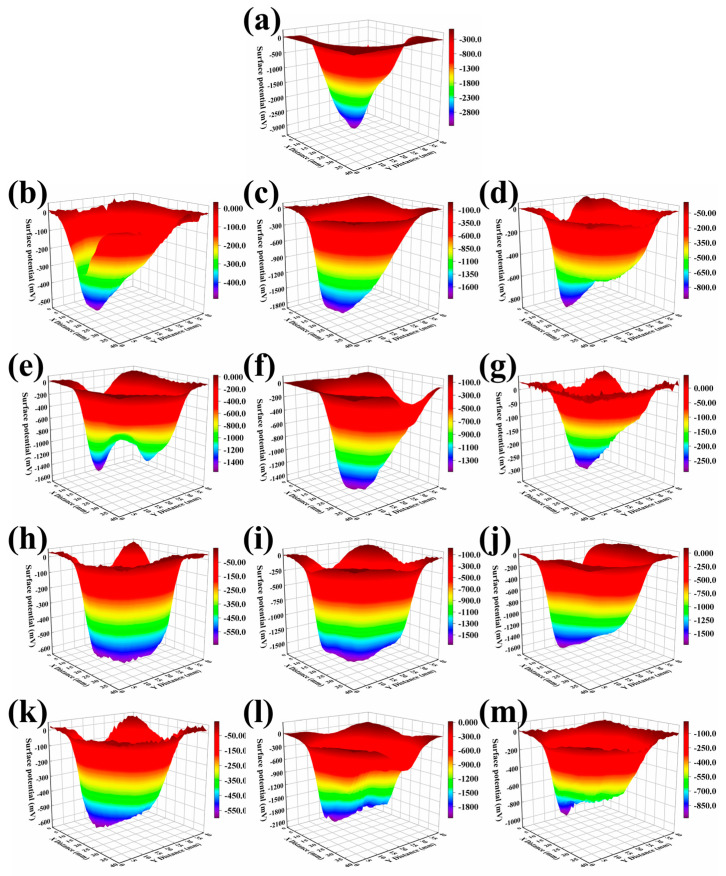
The surface potential test results. (**a**) Pure EP; (**b**) 0.1% GNSs/EP; (**c**) 0.2% GNSs/EP; (**d**) 0.3% GNSs/EP; (**e**) 0.4% GNSs/EP; (**f**) 0.5% GNSs/EP; (**g**) 0.6% GNSs/EP; (**h**) 0.1% DFGNSs/EP; (**i**) 0.2% DFGNSs/EP; (**j**) 0.3% DFGNSs/EP; (**k**) 0.4% DFGNSs/EP; (**l**) 0.5% DFGNSs/EP; (**m**) 0.6% DFGNSs/EP.

**Figure 10 nanomaterials-14-02009-f010:**
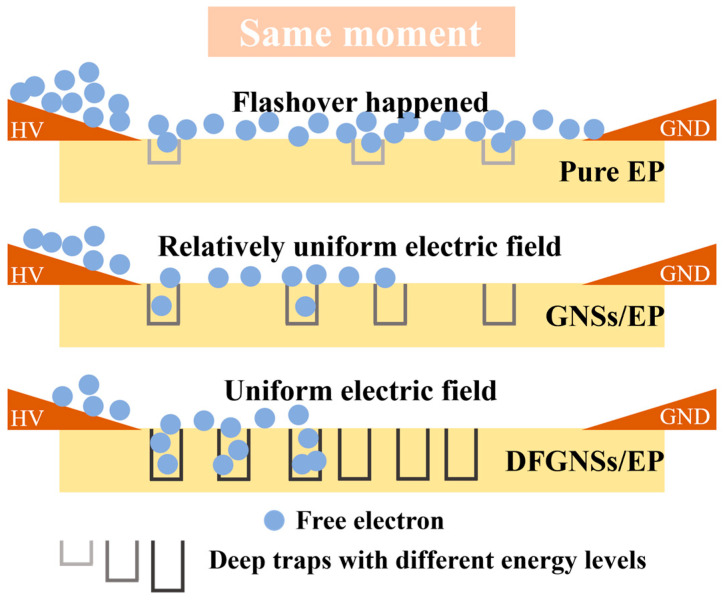
Schematic diagram of enhancement mechanism.

## Data Availability

Data will be provided upon request; please contact us if necessary. The email is as follows: xiaguoweihd@163.com.

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
