# Peer review of "Enhance the Surface Insulation Properties of EP Materials via Plasma and Fluorine-Containing Coupling Agent Co-Fluorinated Graphene"

_nanomaterials, 2024, doi:10.3390/nano14242009_

Round 1

Reviewer 1 Report

Comments and Suggestions for Authors

The authors present an innovative study focused on enhancing the surface insulation properties of epoxy resin (EP) composites by incorporating plasma and fluorine-containing coupling agent co-fluorinated graphene nanosheets (DFGNSs). The study utilizes comprehensive characterization and testing to demonstrate that DFGNSs significantly improve insulation properties at optimized concentrations, achieving a 16.23% increase in flashover voltage at 0.2wt%. While the study is methodically conducted and offers valuable insights, several issues need to be addressed before it can be recommended for publication. Detailed comments are as follows:

  1. Clearer and more detailed notations should be added to Figure 1 to improve reader understanding.
  2. Including photos of treated and untreated samples would provide a better visual comparison and enhance the clarity of the results.
  3. How many trials were conducted for the flashover tests? Providing an error bar is essential to ensure statistically meaningful results.
  4. The labels in Figure 9 are too small and difficult to read. Improving their size and clarity is necessary.
  5. Additional information about the electric field distribution in the testing setup is required to better interpret the results. Simulation is encouraged. 

Author Response

Comments 1: Clearer and more detailed notations should be added to Figure 1 to improve reader understanding.

Response 1: We thank the reviewer for the suggestions. Your statement is absolutely correct. We have modified Figure 1 to provide a better reading experience for the reader. The relevant changes can be found exactly on page 3, line 128.

Comments 2: Including photos of treated and untreated samples would provide a better visual comparison and enhance the clarity of the results.

Response 2: We thank the reviewer for the suggestions. For the surface topography of treated and untreated samples, we also conducted relevant tests, as shown in the figure below. However, since the modification method of epoxy resin is bulk doping modification, this modification method has little influence on the surface morphology of the sample. In addition, the main innovation of this paper is the functionalization of graphene to improve the properties of composite materials, so this result is not included in the paper.

Comments 3: How many trials were conducted for the flashover tests? Providing an error bar is essential to ensure statistically meaningful results.

Response 3: We thank the reviewer for the suggestions. We carried out 10 flashover tests on each sample respectively. Due to our negligence, error bars were not added to the results to reflect the error of the results. Now the error bar has been added to the result chart, and the full text has been checked and verified to avoid the recurrence of similar problems. The relevant changes can be found precisely on lines 230 and 232 of page 7.

Comments 4: The labels in Figure 9 are too small and difficult to read. Improving their size and clarity is necessary.

Response 4: We thank the reviewer for the suggestions. Your statement is absolutely correct. For ease of reading, we have enlarged the labels and scales in Figure 9 to improve their clarity. The relevant changes can be found exactly on page 9, line 289.

Comments 5: Additional information about the electric field distribution in the testing setup is required to better interpret the results. Simulation is encouraged.

Response 5: We thank the reviewer for the suggestions. Your statement is absolutely correct. In the experiment of the surface potential distribution of the sample, we use the needle plate electrode for corona charging. The electric field of this charging method is hemispherical, and the electric field distribution on the sample surface is relatively clear after charging. At the same time, our test results are presented as surface potential distribution, which can further explain the distribution of electric field.

Reviewer 2 Report

Comments and Suggestions for Authors

1.       The phrases "surface insulation properties", “EP materials via plasma”, “coupling agent modification”, “insulation property”, and “co-fluorinated graphene” are crucial for this paper, necessitating the inclusion of key literature in the introduction and relevant conclusions in the conclusion section.

2.       It is essential to include a conclusion about "epoxy resin" in the conclusions section.

3.       It is important to include relevant results about “graphene nanosheets” in the conclusions section.

4.       In the conclusions section, it is essential to include a final statement for each figure and table to clearly outline all relevant points in both a specific and general discussion.

5.       Equations 1, 2, and 3 should include relevant conclusions about the advantages of using these mathematical models and the most meaningful results found with this mathematical tool.

6.       Figure 3 illustrates some functional groups. Do you have any questions regarding the technological applications of these functional groups?, They can be either hypothetical or real.

7.       In Figure 10, the authors need to explain clearly what the filling of the containers represents and the significance of having one, two, or three spheres, or even more.

8.       It is a remarkably interesting paper. My observations aim to enhance readers' interest while always respecting the authors' ideals.

Any work that requires improvement needs the authors to revise the English

Author Response

Comments 1: The phrases "surface insulation properties", “EP materials via plasma”, “coupling agent modification”, “insulation property”, and “co-fluorinated graphene” are crucial for this paper, necessitating the inclusion of key literature in the introduction and relevant conclusions in the conclusion section.

Response 1: We thank the reviewer for the suggestions. We have added the literature on plasma-modified EP, insulation properties of EP, etc., in the introduction and conclusions sections. The relevant changes can be accurately found in lines 56, 57, 58, 59 and 60 on page 2 of the paper.

Comments 2: It is essential to include a conclusion about "epoxy resin" in the conclusions section.

Response 2: We thank the reviewer for the suggestions. Your statement is absolutely correct.

We have added the statement about the insulation properties of epoxy resin in the conclusion. The relevant changes can be found on lines 334, 335, 336, 337 and 338 on page 11.

Comments 3: It is important to include relevant results about “graphene nanosheets” in the conclusions section.

Response 3: We thank the reviewer for the suggestions. In the conclusion part, we added the expression about the microstructure of graphene nanosheets. The relevant changes can be found on lines 328, 329 and 330 on page 11. 

Comments 4: In the conclusions section, it is essential to include a final statement for each figure and table to clearly outline all relevant points in both a specific and general discussion.

Response 4: We thank the reviewer for the suggestions. We have made a final statement of each figure and table in the conclusion section and have summarized the main points. The relevant changes can be found on page 11 of this article on lines 339, 340, 341, 342, 343, and 344.

Comments 5: Equations 1, 2, and 3 should include relevant conclusions about the advantages of using these mathematical models and the most meaningful results found with this mathematical tool.

Response 5: We thank the reviewer for the suggestions. According to the existing research, there is a certain relationship between the surface charge dissipation of materials and the surface trap characteristics. The existence of a trap will constrain the motion of the charge, and the deeper the trap level, the more difficult it is for the charge to escape and dissipate. Conversely, the shallower the trap level, the easier it is for the charge to escape, thus accelerating the charge dissipation. Based on the study of energy band theory, Simmons proposed the isothermal surface potential decay method (ISPD) to calculate the trap characteristics of the material surface, and its main formulas are the formulas (1), (2) and (3) mentioned in the paper.

[1] Yin, K.; Xie, Q.; Ruan, H.O.; Duan, Q.J.; Lu, F.C.; Bian, X.M.; Zhang, T. Causation of ultra-high surface insulation of Bi0.95Y0.05FeO3 /epoxy composites: Simultaneous sine-variations of dielectric and trap properties with filler content. Compos. Sci. Technol. 2020, 197, 108199.

[2] Xie Q.; Duan Q.J.; Xia G.W.; Li J.W.; Yin K., Xie J. Effect of liquid diffusion and segregation on GFRP insulation performance in typical hygrothermal environment. Compos. Part B 2022, 244, 110152.

[3] Wang W.W.; Min D.M.; Li S.T. Understanding the conduction and breakdown properties of polyethylene nanodielectrics: effect of deep traps. IEEE Trans. Dielectr. Electr. Insul. 2016; 23, 564–72.

[4] Simmons J.G.; Tam M. C. Theory of Isothermal currents and the direct determination of trap parameters in semiconductors and insulators containing arbitrary trap distributions. Phys. Rev. B 1973, 7, 3706-3713.

Comments 6: Figure 3 illustrates some functional groups. Do you have any questions regarding the technological applications of these functional groups?, They can be either hypothetical or real.

Response 6: We thank the reviewer for the suggestions. The functional groups are obtained by FTIR test results. FTIR is an instrument used to measure and analyze the infrared spectrum of a sample. It is able to provide information about the chemical bonds and functional groups in the sample, and the infrared spectrum shows the infrared absorption intensity of the sample at different wavelengths (or wave numbers), and these absorption peaks correspond to the vibration patterns of specific chemical bonds and functional groups in the sample. By comparing the standard infrared spectra of known substances, the sample can be qualitatively analyzed to determine the chemical composition present in the sample and thus determine the types of functional groups.

Comments 7: In Figure 10, the authors need to explain clearly what the filling of the containers represents and the significance of having one, two, or three spheres, or even more.

Response 7: We thank the reviewer for the suggestions. Your statement is absolutely correct.

This was an oversight on our part in not labeling the containers and spheres in Figure 10. The sphere represents the free electron and the container represents the energy level of the deep trap. The larger the trap energy level, the more free electrons can be captured in the process of free electron transport, improving the insulation performance of the material, but when the number of trapped free electrons is too large, the electric field distortion will occur at the trap and reduce the insulation performance of the material. The relevant changes can be found exactly on page 10, lines 320 and 321.

Comments 8: It is a remarkably interesting paper. My observations aim to enhance readers' interest while always respecting the authors' ideals.

Response 8: We thank the reviewer for the suggestions. Your statement is absolutely correct.

Thank you very much for your comments, which are very important to the improvement and perfection of the quality of our paper. We also hope that with your help, our paper can become more excellent, so as to be published in this journal.
